# A Novel Method for the Object Detection and Weight Prediction of Chinese Softshell Turtles Based on Computer Vision and Deep Learning

**DOI:** 10.3390/ani14091368

**Published:** 2024-05-01

**Authors:** Yangwen Jin, Xulin Xiao, Yaoqiang Pan, Xinzhao Zhou, Kewei Hu, Hongjun Wang, Xiangjun Zou

**Affiliations:** 1College of Engineering, South China Agricultural University, Guangzhou 510070, China; yangtsejin@stu.scau.edu.cn (Y.J.); 202126410423@stu.scau.edu.cn (X.X.); 20213142022@stu.scau.edu.cn (Y.P.); huck_weeeee@whu.edu.cn (K.H.); 2Foshan-Zhongke Innovation Research Institute of Intelligent Agriculture and Robotics, Foshan 528200, China; zxinzhao@126.com

**Keywords:** Chinese softshell turtle, object detection, image processing, deep learning, weight prediction

## Abstract

**Simple Summary:**

In the sorting process of Chinese softshell turtles, it is necessary to classify them based on their weight and accurately identify their plastron and carapace. This process requires heavy manual labor and complex mechanical processing methods. To improve processing efficiency and reduce costs, this article introduces machine vision technology, and a new image processing method is proposed. This method can estimate the weight of Chinese softshell turtles and accurately locate the positions of their plastron and carapace. The automation level of aquaculture can be greatly enhanced, and hardware costs can be reduced through software optimization through this approach.

**Abstract:**

With the rapid development of the turtle breeding industry in China, the demand for automated turtle sorting is increasing. The automatic sorting of Chinese softshell turtles mainly consists of three parts: visual recognition, weight prediction, and individual sorting. This paper focuses on two aspects, i.e., visual recognition and weight prediction, and a novel method for the object detection and weight prediction of Chinese softshell turtles is proposed. In the individual sorting process, computer vision technology is used to estimate the weight of Chinese softshell turtles and classify them by weight. For the visual recognition of the body parts of Chinese softshell turtles, a color space model is proposed in this paper to separate the turtles from the background effectively. By applying multiple linear regression analysis for modeling, the relationship between the weight and morphological parameters of Chinese softshell turtles is obtained, which can be used to estimate the weight of turtles well. An improved deep learning object detection network is used to extract the features of the plastron and carapace of the Chinese softshell turtles, achieving excellent detection results. The mAP of the improved network reached 96.23%, which can meet the requirements for the accurate identification of the body parts of Chinese softshell turtles.

## 1. Introduction

The Chinese softshell turtle (*Pelodiscus sinensis*), also known as water fish, turtle, or pond fish, belongs to the order *Testudines*, family *Trionychidae*, and genus *Pelodiscus*. Chinese softshell turtles are rich in nutrients and have strong nourishing properties, which are highly favored by consumers. Since the 1990s, China’s turtle breeding industry has experienced rapid growth. By 2019, the annual production had exceeded 320,000 tons [1], forming a sizable and distinctive turtle breeding industry that has also spurred the development of related industries.

The shape and size information of Chinese softshell turtles can intuitively reflect their weight information, which is of great significance to the turtle breeding industry. However, obtaining biological information about turtles solely through manual measurements is highly inefficient and results in significant labor costs. Applying machine vision technology to the detection and identification of the external morphology of turtles can effectively address this issue.

Currently, machine vision methods have been widely used in various areas of aquaculture and agriculture, including species identification [2,3,4,5,6,7,8,9], automated counting [10,11,12], fish behavior recognition [13,14], and freshness detection [15,16]. Scholars both domestically and internationally have conducted extensive research in the field of aquatic machine vision. For instance, D.J. White et al. [3] utilized machine vision technology to achieve a species identification accuracy of up to 99.8% for seven species of flatfish. Zhang Zhiqiang et al. [17] established a model to predict fish mass based on the relationship between the lengths and masses of the head, abdomen, and tail of fish. Pinkiewicz et al. [18] developed an analysis system that uses computer vision to analyze the movement and behavior of fish in aquaculture and can detect fish shapes in video recordings to continuously quantify changes in swimming speed and direction. Yinfeng Hao et al. [14] established a relationship model among fish length, post-tail removal fish body area, and mass to predict fish mass.

However, there is still a lack of research on the object detection and external size measurement of Chinese softshell turtles. This study focuses on extracting image features of turtles by machine vision technology and establishing a weight prediction model for turtles to achieve predictive grading based on their weight. Additionally, deep learning methods are employed to detect Chinese softshell turtles, locate their plastron and carapace, and lay the foundation for the subsequent mass estimation of turtles.

In the individual sorting of Chinese softshell turtles, computer vision technology is utilized to estimate the weight of each turtle, and the turtles are classified based on their weights. Subsequently, through visual recognition, the sorted Chinese softshell turtles are identified. Following these processes, we obtain the coordinates of turtle plastron and carapace. The process of automated sorting is illustrated in Figure 1, but this paper focuses on the parts of visual recognition and weight prediction.

The individual sorting process utilizes images captured by industrial cameras to calculate the morphological parameters of Chinese softshell turtles. Based on the fitted relationship between the parameters and mass, the mass of the turtles is estimated, and sorting is conducted according to their mass. The workflow of mass estimation is illustrated in Figure 2.

For the purpose of visual recognition and weight estimation, we use image processing algorithms to separate the turtles from the background, calculate the rotation angle, and restore the turtle to the standard state; then, we use deep learning detection to detect the turtle, calculate the pixel length, and then convert it to the real length. Finally, the true length and mass prediction model of plastron and carapace is used to estimate the weight of the turtles.

In the visual recognition process, a deep learning algorithm is employed to accurately locate the plastron and carapace of Chinese softshell turtles. The coordinates of plastron and carapace are then transmitted to the subsequent weight prediction part. The workflow of visual recognition is depicted in Figure 3.

The main focus of this paper is on the visual algorithms used in the sorting and visual recognition processes. We propose a color model to separate individual Chinese softshell turtles from the background in images. Additionally, we utilize multivariate linear regression analysis to fit the relationship between the weight and morphological parameters of Chinese softshell turtles and identify suitable models to estimate their weight. Furthermore, we improve the YOLOv7 deep learning object detection network, resulting in a significant increase in detection accuracy for the plastron and carapace of Chinese softshell turtles.

In object detection based on deep learning, the attention mechanism, which is a technology that imitates cognitive attention, [19] is a very important method. The attention mechanism can enhance the weight of certain parts of the input data in the network while weakening the weight of other parts, thereby achieving the purpose of allowing the entire neural network to focus on the place where it needs the most attention, and this is an adaptive process. This article introduces two attention mechanisms, SE attention and SimAM, to improve YOLOv7, and the improved network is named YOLOv7-SS, which comprises YOLOv7, SE, and SimAM.

## 2. Materials and Methods

### 2.1. Experimental Materials and Platform Setup

In this experiment, a total of 153 Chinese softshell turtles were selected, all of which were male individuals with body lengths ranging from 153.9 to 221.6 mm and body weights ranging from 388.9 to 1086.4 g. All Chinese softshell turtles used in this experiment were obtained from a Chinese softshell turtle breeding farm in Guangdong Province.

A “one picture, one turtle” scenario was designed by using a 68 × 130 cm blue PVC background board to capture the image dataset of Chinese softshell turtles.

The image acquisition platform is shown in Figure 4. The binocular camera was fixed on a tripod and transmits captured images to the computer via a USB data cable. The resolution of the binocular camera was 1280 × 960 pixels, and the baseline length was 6 cm. The measurement algorithm was implemented in C++ programming language, utilizing the OpenCV computer vision library for image-related operations and the PyTorch framework for building and training deep learning models.

Prior to capturing images, the external parameters of each Chinese softshell turtle were measured. The required morphological parameters include weight (g), plastron length (cm), plastron width (cm), plastron total length (cm), carapace length (cm), and carapace width (cm), as detailed in Table 1. A schematic diagram of the parameters is shown in Figure 5. For each Chinese softshell turtle, external parameters were measured by using a vernier caliper with a precision of 0.1 mm, and an electronic scale with a precision of 0.01 g was used for weight measurement. A total of 153 male Chinese softshell turtles were measured for external parameters, and over 11,000 images were collected.

### 2.2. Image Processing

Image processing involves several key steps, including image preprocessing, object detection, region of interest (ROI) extraction, and feature extraction.

The experiments in this paper were based on the PyTorch 1.9.0 deep learning framework, running on the Windows 10 operating system. Regarding hardware, we mainly used Nvidia GeForce RTX 2060 GPU to complete the training of the deep learning object detection network. Table 2 shows detailed information about the specific experimental environment configuration.

#### 2.2.1. Image Preprocessing

Since the preprocessing of the captured images is required, a new color model is proposed for segmenting the targets. This model can be represented as follows:(1)$$\left\{\begin{array}{c} f\left(i,j\right)=C_{r}\cdot R\left(i,j\right)+C_{g}\cdot G\left(i,j\right)+C_{b}\cdot B\left(i,j\right)\\ C_{r}+C_{g}+C_{b}=0\\ |C_{r}\left|+\right|C_{g}\left|+\right|C_{b}|>1 \end{array}\right.$$

In this color model, $C_{r}$, $C_{g}$, and $C_{b}$ represent the coefficients of the *R*, *G*, and *B* channels, respectively, while *R*, *G*, and *B* denote the pixel values of the *R*, *G*, and *B* channels in the image. The region composed of points that satisfy this model is considered the target to be detected.

To convert the image to grayscale based on the above color model, the following formula is used:(2)$$f\left(i,j\right)=\left\{\begin{array}{cc} 255, & I\leq Thresh\\ 0, & I>Thresh \end{array}\right.$$
where f(i,j) represents the pixel value at coordinates (i,j)

In the preprocessing of the images of the back of the Chinese softshell turtles, since a blue background was used in this experiment, the parameters used in this paper were $C_{r}=-1$, $C_{g}=-1$, $C_{b}=2$, and Thresh=70.

After preprocessing with this color model, applying a closing operation to the processed result can completely separate the individual Chinese softshell turtle from the background.

#### 2.2.2. Contour Extraction

An algorithm proposed by S. Suzuki et al. [20] was employed in this study to perform topological analysis on binary images and extract the contours of the Chinese softshell turtles.

This algorithm utilizes encoding to assign different integer values to different boundaries, allowing for the determination of boundary connections and hierarchical relationships. The input binary image consists of pixel values of 0 and 1, denoted by f(i,j). The algorithm terminates scanning in the two following cases:(1)f(i,j−1)=0, f(i,j)=1, where f(i,j) is the starting point of the outer boundary.(2)f(i,j)≥0, f(i,j+1)=0, where f(i,j) is the starting point of the hole boundary.

Then, starting from the start point, the algorithm marks the pixels on the boundary. A unique identifier, referred to as NBD (New Boundary Detection), is assigned to each newly discovered boundary. Initially, NBD=1, and it is incremented by 1 each time a new boundary is discovered. During this process, if f(p,q)=1 and f(p,q+1)=0, f(p,q) is set to −NBD. The extracted contours from this step are used for further processing.

#### 2.2.3. Pose Estimation

The moments in the image [21,22] are defined as follows:(3)$$M_{00}=\sum_{I}i*V\left(i,j\right)$$
where $M_{00}$ is a moment of order 0, the image here is a single-channel image, and V(i,j) represents the gray value of the image at point (i,j).
(4)$$M_{10}=\sum_{I}\sum_{J}i*V\left(i,j\right)$$
(5)$$M_{01}=\sum_{I}\sum_{J}j*V\left(i,j\right)$$
where $M_{10}$ and $M_{01}$ are both first-order moments, and *i* and *j* represent the horizontal and vertical coordinates of the image.

When the image is a binary graph, it can be used to calculate the center of gravity of the binary image. The formula is as follows:(6)$$x_{c}=\frac{M_{10}}{M_{00}},y_{c}=\frac{M_{01}}{M_{00}}$$
where $x_{c}$ represents the horizontal coordinate of the target’s center of gravity and $y_{c}$ represents the vertical coordinate of the target’s center of gravity.
(7)$$M_{20}=\sum_{I}\sum_{J}i^{2}*V\left(i,j\right)$$
(8)$$M_{02}=\sum_{I}\sum_{J}j^{2}*V\left(i,j\right)$$
(9)$$M_{11}=\sum_{I}\sum_{J}i*j*V\left(i,j\right)$$
where $M_{20}$, $M_{11}$, and $M_{02}$ represent the second moments.

In an image, these second moments can be used to calculate the orientation of objects. The formula is as follows:(10)$$\theta =\frac{1}{2}arctan\left(\frac{2b}{a-c}\right)$$
where $a=\frac{M_{20}}{M_{00}}-x_{c}^{2},b=\frac{M_{11}}{M_{00}}-x_{c}y_{c},c=\frac{M_{02}}{M_{00}}-y_{c}^{2}$, and θ represents the rotation angle.

We perform moment calculation on all the coordinates of the extracted contours in the image; then, the angle of the Chinese softshell turtles can be calculated according to Equation (Equation 10).

### 2.3. Mass Prediction Model

In this paper, the morphological parameters of 153 male Chinese softshell turtles from a breeding farm in Guangdong were measured. The relationship between each parameter and the mass was statistically analyzed, and a mass prediction model for Chinese softshell turtles was established. The linear regression models between the mass of Chinese softshell turtles and various morphological parameters were built by using SPSS 26.0 software, based on which the mass of Chinese softshell turtles was predicted.

The evaluation metrics for the regression models include $R^{2}$ score, mean absolute error (MAE), mean square error (MSE), and root mean square error (RMSE), defined as follows:(11)$$R^{2}=1-\frac{\sum_{i=1}^{n}{\left({\hat{y}}_{i}-\bar{y}\right)}^{2}}{\sum_{i=1}^{n}{\left(y_{i}-\bar{y}\right)}^{2}}$$
(12)$$MAE=\frac{\sum_{i=1}^{n}\left|{\hat{y}}_{i}-\bar{y}\right|}{n}$$
(13)$$MSE=\frac{\sum_{i=1}^{n}{\left({\hat{y}}_{i}-\bar{y}\right)}^{2}}{n}$$
(14)$$RMSE=\sqrt{\frac{\sum_{i=1}^{n}{\left({\hat{y}}_{i}-\bar{y}\right)}^{2}}{n}}$$
where ${\hat{y}}_{i}$ represents the predicted mass, $y_{i}$ denotes the actual mass, $\bar{y}$ represents the mean, and n is the sample size.

### 2.4. Object Detection Algorithm YOLOv7-SS

Figure 6 depicts the images of Chinese softshell turtles collected in this experiment, where Figure 6a shows the plastron and Figure 6b shows the carapace.

After calculating the pose of the Chinese softshell turtle, the region of interest (ROI) is extracted, and the turtle is rotated to a standard position based on the rotation angle, with the head horizontally oriented to the left. In this paper, this operation is referred to as standardization of the target. At this point, the Chinese softshell turtle is considered to be in a standard state. Figure 7 shows the extracted region of interest (ROI).

Due to the significant variations in the positions of the limbs, head, and tail of the Chinese softshell turtle across different images, traditional image segmentation algorithms face challenges in segmenting turtles in images. Therefore, this paper adopts a deep learning-based approach for object detection.

Since the introduction of the You Only Look Once (YOLO) algorithm by Redmon et al. [23], the field of object detection has made significant progress. The subsequent development of YOLOv2 [24] by the same team further optimized the neural network structure. Additionally, Bochkovskiy et al. proposed the classic YOLOv4 algorithm [25], enabling excellent results to be achieved with a single GPU.

In this paper, we utilize the YOLOv7 algorithm proposed by Wang et al. [26] to train the network on the standardized images of the plastron and carapace of Chinese softshell turtles. Subsequently, we validate the detection results by using non-standardized images of the turtles.

YOLOv7 is a rapid object detection algorithm that was enhanced in this study by incorporating the Squeeze-and-Excitation (SE) attention mechanism [27] and the Simultaneous Attention Mechanism (SimAM) [28].

The SE (Squeeze-and-Excitation) attention mechanism consists of two main steps: Squeeze and Excitation. In the Squeeze step, the feature map undergoes global average pooling to compress the input feature map into a vector. Subsequently, a fully connected layer maps this vector to a smaller 1 × 1 × C vector. In the Excitation step, the elements of the 1 × 1 × C vector are compressed to values between 0 and 1 by using the sigmoid function. This vector is then multiplied with the original input feature map to obtain the weighted feature map. Deep learning network models can utilize the SE attention mechanism to adaptively learn the weights of each channel, thereby enhancing the performance of the model. The schematic diagram of the SE mechanism is illustrated in Figure 8.

The proposal of the SimAM is based on discoveries in the field of neuroscience. It defines the following energy function for each neuron in a neural network:(15)$$e_{t}\left(w_{t},b_{t},y,x_{i}\right)=\frac{1}{M-1}\sum_{i=1}^{M-1}{\left(y_{o}-{\hat{x}}_{i}\right)}^{2}+{\left(1-\left(w_{t}t+b_{t}\right)\right)}^{2}+\lambda w_{t}^{2}$$

The variables $w_{t}$ and $b_{t}$ in the above equation can be obtained by solving the following formula:(16)$$w_{t}=-\frac{2\left(t-\mu_{t}\right)}{{\left(t-\mu_{t}\right)}^{2}+2\sigma^{2}+2\lambda }$$
(17)$$b_{t}=-\frac{1}{2}\left(t+\mu_{t}\right)w_{t}$$
where $\mu_{t}=\frac{1}{M-1}\sum_{i=1}^{M-1}x_{i}\;,\;\sigma_{t}^{2}=\frac{1}{M-1}\sum_{i}^{M-1}{\left(x_{t}-\mu_{t}\right)}^{2}$.

The minimum energy can be calculated by using the following formula:(18)$$e_{t}^{*}=\frac{4\left({\hat{\sigma }}^{2}+\lambda \right)}{{\left(t-\hat{\mu }\right)}^{2}+2{\hat{\sigma }}^{2}+2\lambda }$$

According to the definition of attention mechanism, the input features are enhanced according to the following formula:(19)$$\tilde{X}=sigmod\left(\frac{1}{E}\right)⊙X$$

This yields the formula for the SimAM. Additionally, the neural network in this paper utilizes the Focal GIoU loss function, which combines Focal Loss [29] and GIoU Loss [30].

Focal Loss is an improved loss function based on the cross-entropy loss function. It incorporates a balancing factor, α, to address the issue of imbalanced proportions between positive and negative samples. The formula is as follows:(20)$$L_{fl}=\left\{\begin{array}{cc} \;-\alpha {\left(1-y^{\prime }\right)}^{\gamma }\log y^{\prime } & y=1\\ -\left(1-\alpha \right)y^{\prime^{\gamma }}\log\left(1-y^{\prime }\right) & y=0 \end{array}\right.$$

The ordinary Intersection over Union (IoU) metric struggles to accurately reflect how the predicted box and the ground truth box intersect. Generalized IoU (GIoU) introduces the minimum enclosing rectangle for both the predicted and ground truth boxes to obtain the proportion of overlap between the predicted and ground truth boxes within the enclosing region. Therefore, GIoU not only considers the overlapping region between the two boxes but also pays attention to other non-overlapping regions. GIoU can effectively reflect the intersection of these two boxes within the enclosing region. Its formulation is as follows:(21)$$GIoU=IoU-\frac{\left|C-\left(A\cup B\right)\right|}{\left|C\right|}$$

In the original network architecture, adding attention mechanisms may affect the weights in the original backbone network. Therefore, in this paper, the SimAM is added at the connection between the ELAN layer and the next layer of the original network to minimize the impact on feature extraction. Additionally, the SE attention mechanism is added to the detection head, positioned at the connection between the original ELAN-H layer and the next layer.

After adding the above structures, the modified network architecture is shown in Figure 9. This improved version of YOLOv7 is referred to as YOLOv7-SS in this paper.

When comparing the improved network and the unimproved network, the same hyperparameters are set for each network to prevent differences in hyperparameters from affecting the training results. The hyperparameter settings of the network are shown in Table 3.

#### Parameter Measurement

By using LabelImg software, the images of Chinese softshell turtles collected were annotated. There were over 11,000 images in total, of which the training set images accounted for 80%, the verification set accounted for 10%, and the test set accounted for 10%. The YOLOv7-SS model was trained for 300 epochs. The objects annotated included individual Chinese softshell turtles, their abdomens, and tails. After calculating the rotation angle (θ) of the Chinese softshell turtles and restoring it to the standard position, the images were subjected to object detection. The centroid of the detection box was considered the position of the Chinese softshell turtles.

Then, YOLOv7-SS was used to detect the turtles that had been converted to the standard state. The pixel length of the detection frame can be converted into the actual length of the plastron and carapace of the Chinese softshell turtles. The conversion formula is as follows:(22)$$\begin{array}{cc} s & =\frac{L_{b}}{L_{bp}}\\ L & =s\times L_{p} \end{array}$$
where *s* represents the scale factor, $L_{b}$ is the real length of the measuring tool, and the unit is mm; $L_{bp}$ is the pixel length of the measuring tool in the image, in pixels; *L* represents the real size of the turtle; and $L_{p}$ represents the pixel length of turtle in the image.

## 3. Results

### 3.1. Image Processing Results

The images of a Chinese softshell turtle processed by the polarized color model proposed in this paper are shown in Figure 10, revealing clear outlines of the turtle.

After processing the images by using the polarized color model proposed in this paper, the preliminary segmentation between the target and background is achieved, as shown in Figure 10a. Subsequently, after performing a closing operation in image processing, the individual targets are clearly separated, as depicted in Figure 10b. Then, contour extraction algorithms are applied to extract the outlines of the Chinese softshell turtle, as shown in Figure 11.

After obtaining the coordinates of all contour points, the rotation angle of the target in Figure 11 is calculated to be 139.11° by using Equation (Equation 10). This angle represents the counterclockwise rotation from the positive x-axis direction around the origin. By using this angle, the Chinese softshell turtle is restored to the standard orientation, as shown in Figure 12, before the target detection operation is conducted.

### 3.2. Results of the Weight Model

The comparison results of the quality prediction models for the Chinese softshell turtle are presented in Table 4. The table indicates a positive correlation between the turtle’s quality and morphological parameters, with a relatively high degree of correlation.

Specifically, by utilizing plastron length ($L_{P}$), plastron width ($W_{P}$), and full length ($L_{F}$) as independent variables and quality as the dependent variable, the quality model exhibits a high degree of fit, with an R-squared value of 0.916. Moreover, the maximum relative error (MaxRE) among the three models is minimal for this configuration, only 9.67%.

### 3.3. Object Detection Results

The improved YOLOv7-SS object detection network can accurately identify the plastron and carapace of the Chinese softshell turtle, as shown in Figure 13.

To clearly examine the impact of the improvements in the algorithm, this study conducted comparative experiments on the test set of Chinese softshell turtle images. YOLOv5, YOLOv7, and the proposed YOLOv7-SS were compared. The comparative results are presented in Table 5.

The precision of YOLOv7-SS is 95.28%, which is nearly 8% higher than the original YOLOv7 and 13.42% higher than the YOLOv5 algorithm, indicating a significant performance improvement. The comparison of mAP values between YOLOv7-SS and the original YOLOv7 is illustrated in Figure 14.

## 4. Discussion

Computer-assisted digital image processing is widely used in animal weight estimation. For example, C.P. Schofield [31] applied image analysis techniques to estimate the weight of pigs. Sirimonpak et al. [32] estimated the weight of pigs by extracting and calculating data such as the lengths of the major and minor axes, centroid, and eccentricity. Similar methods have also been applied to weight estimation in rabbits [33], broilers [34], and other animals. In this article, we propose a polar model image processing method and, for the first time, estimate the weight of Chinese softshell turtles. This method shows good estimation performance in the weight estimation of Chinese softshell turtles.

Although the color model we proposed still shows some noise after image processing, the expected effect can be achieved through subsequent image closing operations. In general, the segmentation method based on color space proposed in this article can perfectly segment the target from the background.

Multiple linear regression analysis is used to explore the relationship between the weight and morphological parameters of Chinese softshell turtles for weight estimation. Although the R-squared value of the multiple linear regression model is only 0.916, it represents a big step forward in exploring the relationship between the morphological parameters and center of gravity of Chinese softshell turtles, laying the foundation for the subsequent accurate weight estimation of Chinese softshell turtles.

Generally speaking, when the number of samples participating in multiple linear regression is larger, the calculated regression model is more accurate. Therefore, it should be feasible to improve the accuracy of the regression model by increasing the number of Chinese softshell turtles. Our subsequent work will consider this question.

The detection accuracy of YOLOv7-SS is 8% higher than that of the original YOLOv7, and the convergence speed is also very fast. However, there is a jitter problem during the training process, which is not as smooth as the convergence process of the original YOLOv7. The detection and weight prediction method for Chinese softshell turtles proposed in this article has completed the first two steps of turtle sorting and laid the foundation for the automated sorting of turtles.

Although the method proposed in this article has good detection results for a single turtle, in the scenario of multiple turtles, there will be mutual occlusion, which seriously affects the weight estimation of the visual recognition system. The goal of our subsequent work is to solve this problem.

## 5. Conclusions

The proposed polarized model effectively separates individual Chinese softshell turtles from the background, demonstrating clear segmentation results.

The results of multiple linear regression analysis indicate a certain linear relationship between the weight of Chinese softshell turtles and their morphological parameters. Specifically, the variables plastron length ($L_{P}$), plastron width ($W_{P}$), and full length ($L_{F}$) serve as independent variables, while weight serves as the dependent variable. The obtained model exhibits a high degree of fit, with an $R^{2}$ value of 0.916. However, there is still room for improvement in the accuracy of the fitting model. Future work should focus on enhancing the precision of the fitting model to better predict the weight of Chinese softshell turtles.

The improved YOLOv7-SS algorithm shows a significant increase in detection accuracy for Chinese softshell turtles. Although the mAP value of YOLOv7-SS fluctuates considerably during training, it eventually converges to a satisfactory result. Future efforts will explore methods to enhance convergence speed and stability during training.

In summary, our main contributions are reported below.

(1)A color space model is proposed to separate individual Chinese softshell turtles from the background effectively.(2)Multiple linear regression analysis is used to explore the relationship between the weight and morphological parameters of Chinese softshell turtles for weight estimation, which shows a high degree of fit, with an $R^{2}$ value of 0.916.(3)YOLOv7-SS, an improved YOLOv7 deep learning object detection network that includes the SE attention mechanism and SimAM, is used to extract the features of the plastron and carapace of Chinese softshell turtles, and its detection accuracy can reach 96.23%, which is nearly 8% higher than that of the original YOLOv7.

## Figures and Tables

**Figure 1 animals-14-01368-f001:**
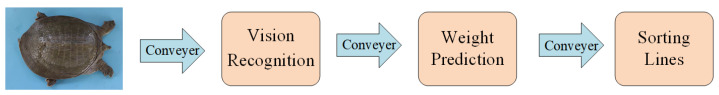
Turtle-sorting flow chart.

**Figure 2 animals-14-01368-f002:**
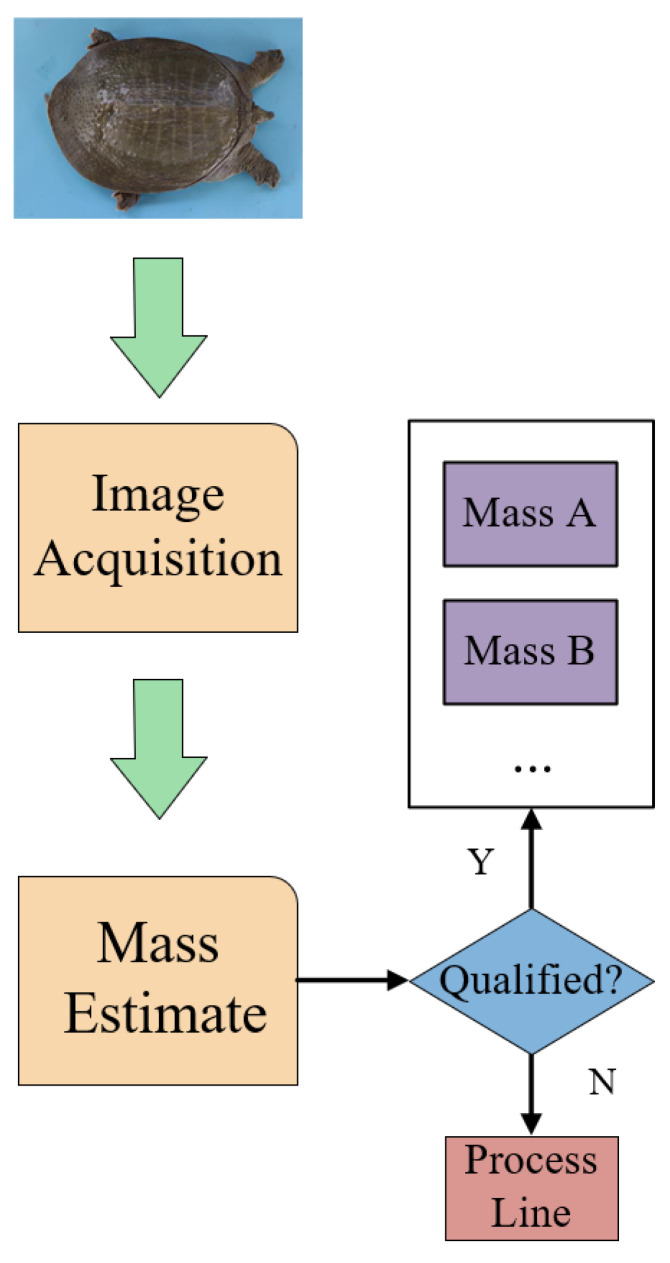
Mass estimation flow chart.

**Figure 3 animals-14-01368-f003:**
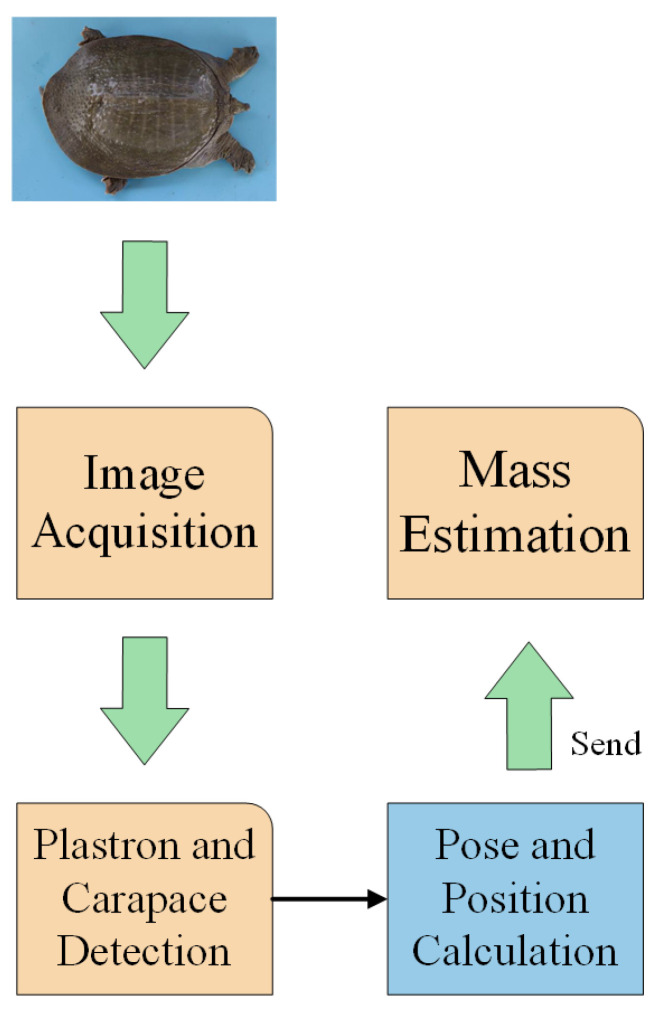
Visual recognition flow chart.

**Figure 4 animals-14-01368-f004:**
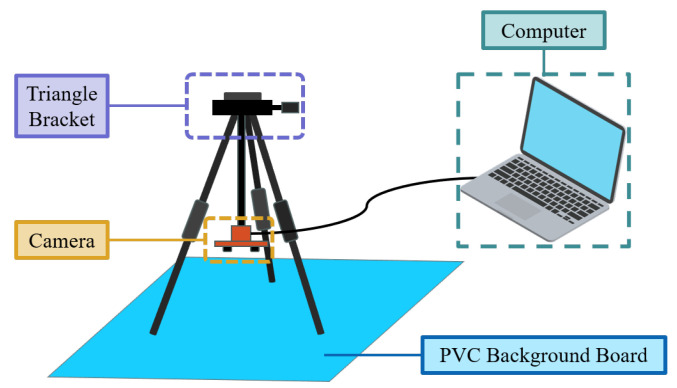
Image acquisition platform.

**Figure 5 animals-14-01368-f005:**
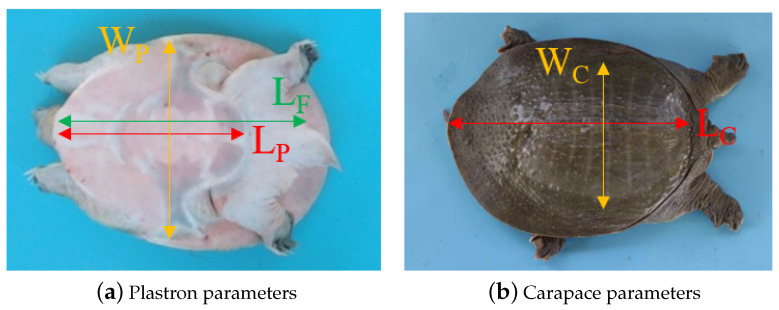
Morphological parameters of *Pelodiscus sinensis*.

**Figure 6 animals-14-01368-f006:**
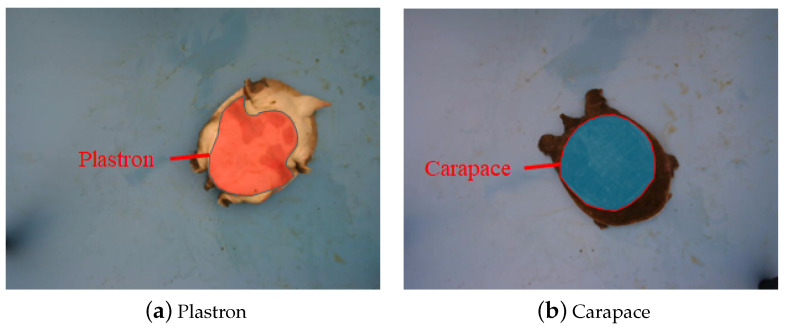
Sample images of Chinese softshell turtle.

**Figure 7 animals-14-01368-f007:**
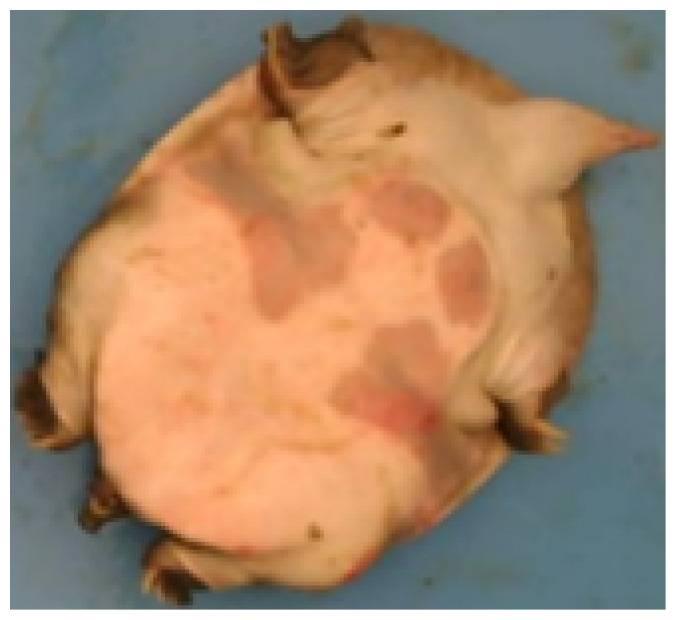
Region of interest.

**Figure 8 animals-14-01368-f008:**
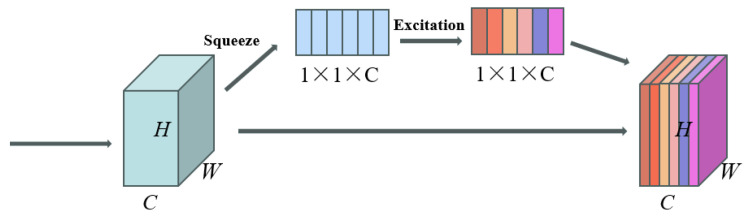
Principle of SE attention. Here, *C* represents the number of channels in the image, typically 3, indicating that the image is an RGB color image; *H* represents the height of the image, i.e., the vertical size of the image, usually measured in pixels; *W* represents the width of the image, i.e., the horizontal size of the image, also typically measured in pixels.

**Figure 9 animals-14-01368-f009:**
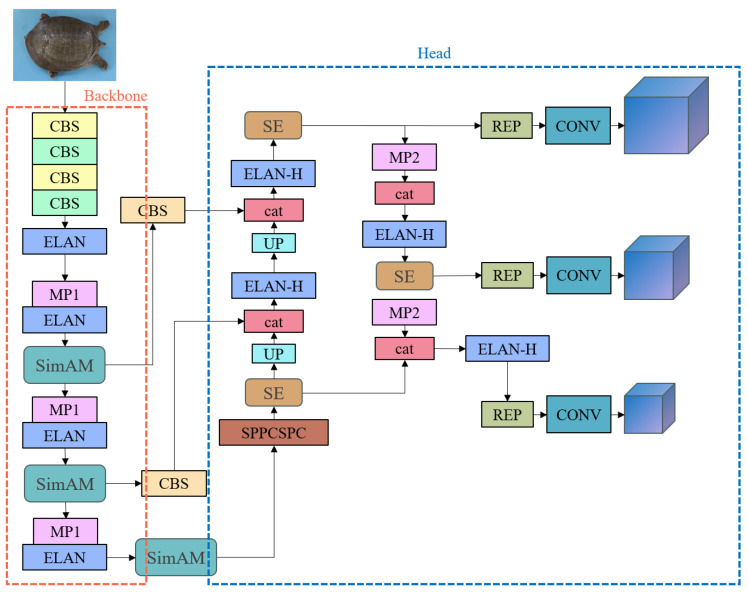
The structure of the YOLOv7-SS network.

**Figure 10 animals-14-01368-f010:**
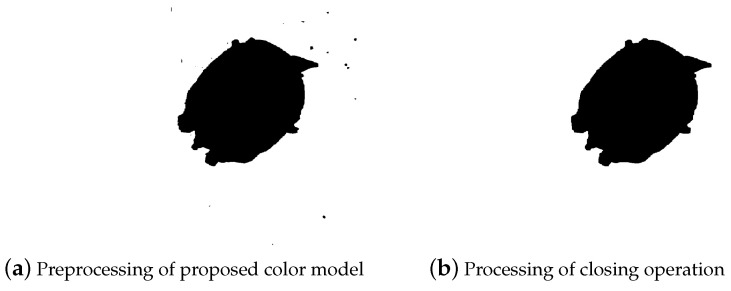
Image processing results.

**Figure 11 animals-14-01368-f011:**
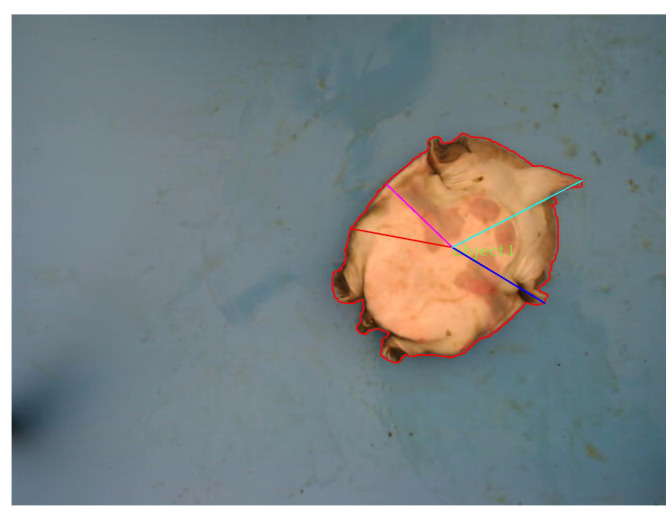
Extracted contour.

**Figure 12 animals-14-01368-f012:**
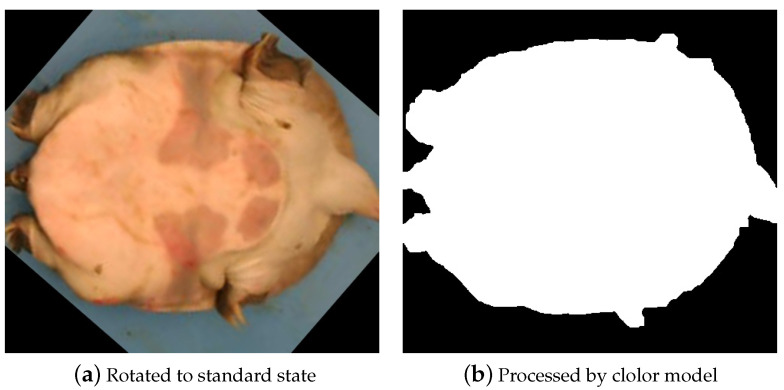
Standard state.

**Figure 13 animals-14-01368-f013:**
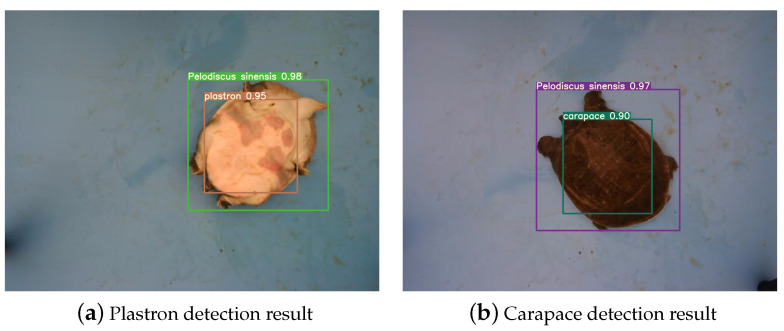
Object detection results of Chinese softshell turtle.

**Figure 14 animals-14-01368-f014:**
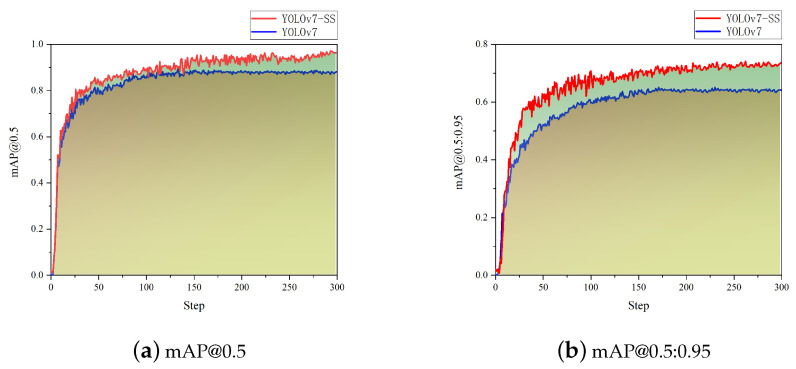
mAP comparison.

**Table 1 animals-14-01368-t001:** The description and definition of the morphological parameters of the Chinese softshell turtles.

Morphological Parameter	Definition
Mass (*M*)	The mass of the Chinese softshell turtle (g)
Carapace length ($L_{C}$)	The maximum straight-line distance from the anterior to the posterior end of the carapace
Carapace width ($W_{C}$)	The maximum straight-line distance from the left side to the right side of the carapace
Plastron full length ($L_{F}$)	The straight-line distance from the anterior end of the plastron to the beginning of the tail
Plastron length ($L_{P}$)	The maximum straight-line distance from the anterior to the posterior end of the plastron
Plastron width ($W_{P}$)	The maximum straight-line distance from the left side to the right side of the plastron

**Table 2 animals-14-01368-t002:** Experimental environment.

Configuration	Parameters
Operating System	Windows 10
CPU	Intel(R) Core(TM) i5-10400F @2.9 GHz
GPU	Nvidia GeForce RTX 2060
Deep learning environment	CUDA 11.2, CUDNN 8.1.1.33, and Pytorch 1.9.0
Image library	OpenCV 4.5.3
Development tools	Visual Studio 2019

**Table 3 animals-14-01368-t003:** The hyperparameters in the training process.

Parameter	Value
epoch	300
initial learning rate	0.01
batch size	8
momentum	0.937
weight decay	0.0005
box	0.05
cls	0.3
obj	0.7

**Table 4 animals-14-01368-t004:** Comparison of Chinese softshell turtle mass prediction models.

Dependent Variable	Predictor Variables	$R^{2}$	MAE (g)	RMSE (g)	MaxRE (%)
Mass	$L_{C}+W_{C}$	0.883	42.08	47.38	12.39
	$L_{P}+W_{P}$	0.903	38.13	44.97	11.58
	$L_{P}+W_{P}+L_{F}$	0.916	35.33	40.28	9.67

**Table 5 animals-14-01368-t005:** Comparison of performance of different object detection networks.

Detection Algorithm	Precision (P)	Recall (R)	mAP@0.5	mAP@0.5:0.95
YOLOv5	81.96%	90.52%	89.82%	56.15%
YOLOv7	87.41%	92.74%	88.10%	64.22%
YOLOv7-SS	95.38%	94.68%	96.23%	73.63%

## Data Availability

The data presented in this study are available on request from the corresponding author.

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
