# Peer review of "A Novel Method for the Object Detection and Weight Prediction of Chinese Softshell Turtles Based on Computer Vision and Deep Learning"

_animals, 2024, doi:10.3390/ani14091368_

Round 1
Reviewer 1 Report
Comments and Suggestions for Authors
Reconsider after major revision -
A major problem with this paper is that there is no discussion of animal welfare problems associated with the cutting points on the plastron and carapace. Citing these points would not quickly produce unconsciousness. To make this paper acceptable for publication will require either a complete discussion about welfare problems or removing ALL references to slaughter. This review recommends removing all statements about slaughter and presenting only the information on sorting and estimating weight of the turtles.
Title - The title of the manuscript is different from the title on the email. This reviewer recommends keeping the new title that emphasizes sorting and weight prediction. The new title has no reference to slaughter.
Line 1 - Remove first sentence on slaughter.
Line 6-7 - Remove information on the cuts.
Line 9-25 - Remove all information about cutting and slaughter and concentrate on using the vision program for sorting and weight prediction.
Lines 71-75 - Remove all the information about cutting.
Line 289 - Discuss if a sample of turtles larger than 153 turtles would improve accuracy.
Comments on the Quality of English LanguageNA
Reviewer 2 Report
Comments and Suggestions for Authors
This paper presented an improved YOLOv7 deep-learning object detection network to extract the features of the abdomen and back of Chinese softshell turtles. In addition, a color space model is proposed to separate individual Chinese softshell turtles from the background effectively and multiple linear regression is used to estimate the weight of Chinese softshell turtles. Some parts of this analysis are not clearly explained and don’t show proper results. I will suggest clarifying your contributions more clearly and giving a proper background of your different analyses. For now, I will leave it for major revision. I have some comments as follows-
1. In the abstract, the authors provided detection accuracy as 96.23%. I will suggest mentioning mAP here.
2. Why is the YOLO-v7 version chosen? The latest YOLO-v8 version could be utilized.
3. Please, explain more detail on the extreme color model.
4. What is YOLOv7-SS? The authors didn’t provide any background information about SE and SimAM attention mechanisms but started to talk about it on page 3. Better to discuss a little before mentioning your contributions summary.
5. What kind of measurement algorithm is used? What is its purpose?
6. Could you please explain if you have multiple turtles in the conveyor system, will your model work? It is better to add a discussion on that.
7. It is not clear how and why you used multiple linear regression for your problem. Could you please explain a little more?
8. Could you please make it more clear whether your problem is segmentation or detection? You talked about segmenting images from the background and doing relevant processing, but you reported mAP results for your localization and classification. It will be better to make it more clear throughout the paper.
Thank you for your good attempts.
Good luck
Reviewer 3 Report
Comments and Suggestions for Authors
The article titled "A novel method on the object detection and weight prediction of Chinese softshell turtle based on computer vision and deep learning" discusses the use of computer vision and deep learning to improve the automated slaughtering process of Chinese soft-shelled turtles. The article presents improvements made through the use of YOLOv7, which achieved a detection accuracy of 96.23%. The authors made modifications to the image processing to enable accurate identification of the body parts of Chinese softshell turtles. While the manuscript contains novel information, some issues were found that require addressing before it can be considered for publication in the journal. Below are the proposed changes that should be made to the manuscript.
The authors should clarify the contributions of the manuscript in the introduction. They should emphasize the issues that arise when taking turtles in the automated process of sacrifice. Although the contributions are described in L77-92 this text would be better placed in the results and conclusions section since there is not enough information in those sections. It is necessary to introduce readers to how the turtle is taken, whether the images are obtained directly from a pond, from a confined place, or taken from the classification lines (processing bands). Additionally, it is necessary to add a section discussing the bioethics involved in the sacrifice of this species and how it is diminished by the use of this technology. The benefits of this new method should not only focus on industry costs (L39), and you are describing (L 12-13& 65-66) the contributions of the sorting and vision recognitions are for better cutting points and by that, better food processing.
Furthermore, I recommend that after the first mention of Chinese softshell turtle an acronym be adopted to avoid the use of so many words throughout the manuscript.
Some paragraphs need to be corrected because the ideas are repeated (L 31). Other paragraphs are too short and may be inappropriate. Authors should organize this content to best present their ideas.
You must define what growth status means (L37) and define the problem in a better way to justify the manuscript, e.g. the unnecessary sacrifice of turtles that do not meet the dimensions, the amount of meat, or some other matter. It would also be interesting to talk in a short paragraph about the fate of the animals' remains once they are sacrificed and why the use of these technologies for more conscious and sustainable processes is important.
Are the processing methods described in the introduction the newest available? Or how is it important to talk about an increase in accuracy? On what basis is this increase described related to the other methods?
In the methodology part, I only doubt how the organisms are placed in the camera to do the image processing. Do the turtles settle down one by one? Are they given time in case they walk? Are they taken out of the pond and put in bands to direct them somewhere, or are they placed somewhere else? Was your process manual or automated with a robotic hand? Is this method they used the one used in turtle farms? in which ones? And why didn't they choose another method? Is this number of turtles representative of the percentage of accuracy they present? At what age were the turtles or is it something described at how many weeks they are removed from the pond for sacrifice?
In summary, more information needs to be given from the introduction so that the methodology only lists what the authors did, why they chose that method, and how it is closer to the reality of current farms. In addition, it is necessary to review the titles of the tables and figures and propose more informative titles.
In the results, the question arose as to why an automated scale is not used when the turtles are passed. There are automated methods of slaughtering tilapia, for example, it weighs the fish and knows how many pieces to put in the package for sale. Shouldn't your image analysis know in advance a close measure of how much Plastron and Carapace weigh based on their age, and diet? Isn't there data from this information that can be added to the model and help obtain a better cut to obtain the meat? Because in the end what they want is for the cuts to be done well to obtain the meat. Is this true? Because I infer that from the paper although it is not said as is. It would be important to clarify what the intention is to obtain a good cut in the slaughter.
Finally, the conclusions do not provide enough information to justify us starting to use this method in the farms for the slaughtering process, it is important to delve deeper into this matter.

The manuscript needs general editing for spelling and writing. In the document, I point out some grammatical errors, missing spaces, and spelling mistakes for your consideration.
Reviewer 4 Report
Comments and Suggestions for Authors
The manuscript entitled by Jin et al., "A novel method on the object detection and weight prediction of Chinese softshell turtle based on computer vision and deep learning" is deals with technic, electronic, informatics more than animal slaughter hygiene and technology
The topic of manuscript in interested and important for improvement of the slaughtering technology of Chinese softshell turtle and aquaculture industry
I recommend to take into consideration the following comments:
Abstract
Line 22: indicated to the full term for the abbreviation (YOLOv7) at first mention
Introduction
Line 54-84: the text is more likely to be belonged to the material and method paragraph, where no any citation of references, otherwise it should be included the scientific citations.
Lines 89-92: the text not belong to the introduction paragraph but its belong to the results paragraph
Materials and Methods
Lines 156-157: Mathematical equation No. 4 is repeated (same as equation No. 5)
Line 158: you indicated that ‘where M10 and M01 are both…’ but M01 not existed in equation No. 4 and 5.
Line 162: you indicated that ‘….. and y represents the vertical coordinate…’ you meaning y or yc?
Figure 8: indicate to the meaning of each: C, H and W
Discussion
The discussion paragraph is very brief which repeated only the results and not include any citation of others related studies.
Reviewer 5 Report
Comments and Suggestions for Authors
-redo the graph in Figure 4 as it appears to be of low quality and not very professional.
-Tab 2 is superfluous; either keep it or retain the information in the text.
-Eqs 1 and 2 are unnecessary; there is no need for a formula to explain the RGB color space or a threshold.
-for the comparison of results between various models and the proposed methods for object detection, it is mandatory to perform hyperparameter optimization using optimizers (e.g., Bayesian optimizers).
-the training technique needs to be better explained, including all the involved hyperparameters.
-Figure 8 can be removed; it is superfluous and well-known in the literature.
-It is suggested to use cross-validation techniques (e.g., k-fold method) due to the limited dataset available.
- For the animal body segmentation, I recommend using more modern techniques, such as SAM (Segment Anything Model), which I believe could work very well and would be less prone to the use of thresholds (like the proposed method).
-It would be beneficial to train all models (segmentation, pose, and weight recognition) and test them on a new set of turtles, never included in the training.
-without hyperparameter optimization, an honest comparison between the various versions of YOLO is not possible, as we cannot determine if the results depend solely on the hyperparameters selected at the time.
Round 2
Reviewer 1 Report
Comments and Suggestions for Authors
Dear Edda Lu - The manuscript ID Animals 2912146 - is now acceptable for publication. The authors have removed all references to slaughter and made the changes I suggested. I approve this paper for publication.
Author Response
Thank you for your kind review and helpful comments.
Reviewer 2 Report
Comments and Suggestions for Authors
I would like to thank all the authors for considering my comments.
I am happy to accept this paper at this point.
Thank you for your contributions.
Good Luck.
Author Response

(The authors gave the same response as above.)

Reviewer 3 Report
Comments and Suggestions for Authors
The authors satisfactorily addressed all comments. There are no additional remarks concerning the other criteria that are given in the manuscript.
Author Response

(The authors gave the same response as above.)

Reviewer 4 Report
Comments and Suggestions for Authors
The authors accepted all my suggestions and comments except include citations of others related studies in the discussion section which is yet without any citation.
Author Response
Thanks for your constructive comments. We've rewritten Simple Summary to meet
your requirements.

Reviewer 5 Report
Comments and Suggestions for Authors
The authors have responded rigorously to all the doubts raised; I find myself in agreement with their statements.
Author Response
Than you for your kind review and useful comments.